# Interpretable machine learning models to predict short-term postoperative outcomes following posterior cervical fusion

**Mert Karabacak**📧, **Konstantinos Margetis**📧*

Department of Neurosurgery, Mount Sinai Health System, New York, New York, United States of America

* Konstantinos.Margetis@mountsinai.org

## Abstract

By predicting short-term postoperative outcomes before surgery, patients who undergo posterior cervical fusion (PCF) surgery may benefit from more precise patient care plans that reduce the likelihood of unfavorable outcomes. We developed machine learning models for predicting short-term postoperative outcomes and incorporate these models into an open-source web application in this study. The American College of Surgeons National Surgical Quality Improvement Program database was used to identify patients who underwent PCF surgery. Prolonged length of stay, non-home discharges, and readmissions were the three outcomes that were investigated. To predict these three outcomes, machine learning models were developed and incorporated into an open access web application. A total of 6277 patients that underwent PCF surgery were included in the analysis. The most accurately predicted outcome in terms of the area under the receiver operating characteristic curve (AUROC) was the non-home discharges with a mean AUROC of 0.812, and the most accurately predicting algorithm in terms of AUROC was the LightGBM algorithm with a mean AUROC of 0.766. The following URL will take users to the open access web application created to provide predictions for individual patients based on their characteristics: https://huggingface.co/spaces/MSHS-Neurosurgery-Research/NSQIP-PCF. Machine learning techniques have a significant potential for predicting postoperative outcomes following PCF surgery. The development of predictive models as clinically useful decision-making tools may significantly improve risk assessment and prognosis as the amount of data in spinal surgery keeps growing. Here, we present predictive models for PCF surgery that are meant to accomplish the aforementioned goals and make them publicly available.

## Introduction

The procedure known as posterior cervical fusion (PCF) is frequently carried out following cervical decompression for conditions such as cervical spondylosis or stenosis, cervical spondylotic myelopathy, ligament ossification, malignancy, or infection [1]. In the United States, the annual rate of PCFs performed for a preoperative diagnosis of degenerative disease climbed 2.7-fold from 2001 to 2013, while the annual number of PCFs performed for a

**Data Availability Statement:** "Restrictions apply to the availability of these data, as the data is shared solely with fellows of American College of Surgeons. Data was obtained from American

College of Surgeons National Surgical Quality Improvement Program and are available (https://www.facs.org/quality-programs/data-and-registries/acs-nsqip/) with the permission of American College of Surgeons."

**Funding:** The authors received no specific funding for this work.

preoperative diagnosis of cervical spondylotic myelopathy increased 2.9-fold from 2003 to 2013 [2, 3]. Increasing number of fellowship-trained spine surgeons, an aging population, increased awareness of quality-of-life issues in this population, improvements in anesthesia, surgical instrumentation and technique, and postoperative care have all contributed to this increase in the annual rates of the procedure [3, 4]. With this increase, it becomes more crucial to select patients who will have successful surgeries in order to maximize their likelihood of success, minimize risks, improve outcomes, and keep costs within control. Furthermore, identifying high-risk patients preoperatively and disclosing those risks to the patients can facilitate shared decision-making and a better-informed consenting process, which ultimately optimizes patient outcomes and satisfaction.

Given the importance of preoperative risk identification and facilitating shared decision-making, the adoption of machine learning (ML) classifiers in clinical prediction models offers a substantial edge over conventional methods. This advantage stems from their capacity to handle complex, high-dimensional, and non-linear relationships among variables [5]. Conventional methods like logistic regression are restricted by their linear nature and require assumptions of variable independence [6]. ML classifiers, such as decision trees, support vector machines, and neural networks, can automatically identify intricate patterns in the data, which often leads to improved predictive accuracy and generalizability [7]. By leveraging the potential of ML prediction models, clinicians can make more informed decisions, ultimately improving patient outcomes and optimizing resource allocation [8].

To our knowledge, there have been no prior studies investigating the ability of ML models to predict prolonged length of stay (LOS), non-home discharges, and 30-day readmissions within one comprehensive study following PCF surgery. The objectives of this research encompass evaluating the effectiveness of ML algorithms for forecasting postoperative outcomes after PCF surgery and developing a user-friendly and readily available tool to achieve this purpose.

## Methods

The methods described below have been also utilized in a different study by the authors of the study [9].

### Ethics statement

This study and the need for informed consent for this study was deemed exempt from approval by the Icahn School of Medicine at Mount Sinai institutional review board because it involved analysis of deidentified patient data.

### Data source

This study utilized data from the American College of Surgeons (ACS) National Surgical Quality Improvement Program (NSQIP) database to identify patients who underwent PCF between 2014 and 2020. Excluding trauma and transplant cases, the ACS-NSQIP database serves as a national surgical registry for adult patients who have had major surgical procedures in various subspecialties at over 700 participating medical centers throughout the United States [10, 11]. Comprehensive information regarding the database and its data collection techniques can be found in other sources [12].

## Guidelines

Transparent Reporting of Multivariable Prediction Models for Individual Prognosis or Diagnosis [13] and Journal of Medical Internet Research Guidelines for Developing and Reporting Machine Learning Predictive Models in Biomedical Research [14] were followed. This was a retrospective machine learning classification study (outcomes were binary categorical) for prognostication in patients who underwent PCF surgery.

## Study population

We queried the NSQIP database to identify patients who met the following inclusion criteria: 1) CPT code 22600 (arthrodesis, posterior or posterolateral technique, single level; cervical below C2 segment), 2) elective surgery, 3) procedure under general anesthesia, and 4) neurosurgery or orthopedics as the surgical subspecialty. We excluded patients based on these criteria: 1) emergency surgery, 2) unclean wounds (wound classes 2 to 4), 3) sepsis/shock/systemic inflammatory response syndrome within 48 hours prior to surgery, and 4) American Society of Anesthesiologists (ASA) physical status classification score of 4, 5 and non-assigned. We also excluded patients who had simultaneous anterior cervical procedures, thoracic and lumbar fusions, revision surgeries, or operations for intraspinal lesions. The CPT codes used to exclude these individuals are listed in S2 Table. We analyzed the ICD-10 codes assigned to patients as the primary diagnoses to further pinpoint those undergoing surgery for degenerative diseases. Patients diagnosed with a fracture, neoplasm, or infection were excluded. Patients asssigned with ICD codes employed fewer than ten times in the entire patient population were likewise excluded to mitigate the effect of rare diagnoses and coding errors.

## Predictor variables

Predictor variables from the NSQIP database, which were assumed to be known preoperatively: 1) demographics like age, sex, race/ethnicity, body mass index [BMI, (determined from height and weight)], and transfer status; 2) comorbidities and disease burden, including diabetes mellitus, current smoker within one year, dyspnea, history of severe chronic obstructive pulmonary disease (COPD), ascites within 30 days before surgery, congestive heart failure (CHF) within 30 days before surgery, hypertension needing medication, acute renal failure, currently requiring or on dialysis, disseminated cancer, steroid or immunosuppressant for a chronic condition, >10% body weight loss in the past 6 months, bleeding disorders, preoperative transfusion of ≥1 unit of whole/packed red blood cells within 72 hours before surgery, ASA classification, and functional status prior to surgery; 3) preoperative lab values like serum sodium, serum blood urea nitrogen (BUN), serum creatinine, serum albumin, total bilirubin, serum glutamic-oxaloacetic transaminase (SGOT), serum alkaline phosphatase (ALP), white blood cell (WBC) count, hematocrit, platelet count, partial thromboplastin time (PTT), International Normalized Ratio (INR) of prothrombin time (PT) values, and PT; 4) operative factors such as surgical specialty, inpatient versus (vs.) outpatient surgical setting, and single vs. multiple-level surgery. CPT code 22614 identified multi-level surgeries. Definitions for these predictor variables can be found in the ACS NSQIP PUF User Guides (https://www.facs.org/quality-programs/data-and-registries/acs-nsqip/participant-use-data-file/). Diabetes variables 'Non-Insulin' and 'Insulin' were combined as 'Yes', and dyspnea variables 'Moderate Exertion' and 'At rest' were also combined as 'Yes'. For transfer status, all variable values other than 'Not transferred (admitted from home)' were categorized as 'Transferred'. Race and ethnicity variables were consolidated into one column, 'Race'. To combine race and ethnicity columns, patients with 'Yes' for 'Hispanic Ethnicity' were re-assigned as 'Hispanic' in the 'Race' column, regardless of their initial 'Race' values.

## Outcomes of interest

The outcomes of interest included prolonged LOS, defined as a total stay exceeding 75% of the studied patient group ($>$ 4 days), discharges to non-home destinations, and readmissions within 30 days. To determine non-home discharges, we categorized the discharge destination variable into two groups. Non-home discharge destinations were identified for patients needing further care after discharge, encompassing 'Rehab', 'Skilled Care, Not Home', 'Separate Acute Care', and 'Multi-level Senior Community'. Patients with unknown discharge destinations, those discharged to hospice care or unskilled facilities, those who left against medical advice, and those who passed away were excluded. Discharges to a 'Facility Which Was Home' were regarded as home discharges.

## Data preprocessing

To avoid introducing bias by excluding patients with missing data, we employed imputation. There were missing values in sixteen continuous variables. We eliminated seven variables that had more than 25% missing values in the patient population and used the nearest neighbor (NN) imputation algorithm to impute missing values for the remaining continuous variables [15]. With NN imputation algorithms, missing values are replaced with values obtained from patients throughout the entire dataset [16]. The only categorical variable with missing values was 'Race', and we filled these missing variables with 'Unknown'.

Continuous variables were scaled using the robust scaler to address outliers [17]. Furthermore, normalization was performed to ensure that all feature values were equally weighted and scaled in the same range. For this purpose, a Min-Max normalization technique was utilized, and each continuous variable, such as BMI or laboratory values, was confined within the [0, 1] range [18]. Categorical variables with ordinal characteristics, like ASA classification and functional status, were encoded using the ordinal encoder [19]. Non-binary categorical variables, such as race and sex, were one-hot-encoded.

The Synthetic Minority Over-sampling Technique (SMOTE) was employed to address class imbalance for a positive outcome of interest by artificially generating instances of positive outcomes [i.e., prolonged length of stay (LOS), non-home discharges, 30-day readmissions] from the training set [20]. To create synthetic instances, SMOTE selects a minority class instance 'a', and identifies its k nearest minority class neighbors. Then, it connects 'a' and 'b' in the feature space to generate a line segment and randomly chooses one of the k nearest neighbors, 'b', connecting it with 'a' to form a synthetic instance. The synthetic instances are produced by combining the selected instances, 'a' and 'b'. [21].

## Training, validation, and test sets

Data from 2014 to 2020 were split into training, validation, and test sets in a 60:20:20 ratio. The models were built using the training set, hyperparameters were optimized using the validation set, and the models' performance was evaluated using the test set.

## Modeling

Four supervised machine learning algorithms, namely XGBoost, LightGBM, CatBoost, and Random Forest, were employed to build models. The Optuna optimization library was utilized to optimize the area under the receiver operating characteristic curve (AUROC). To simplify the process of hyperparameter optimization and make it efficient, Optuna employs various advanced optimization techniques [22]. The Bayesian optimization algorithm Tree-Structured Parzen Estimator Sampler (TPESampler) was used to generate AUROC estimates, which were

used as a guide for the optimization process. Finally, the models were built using the training set and optimized hyperparameters, and the ML analyses were conducted using Python 3.7.15.

## Performance evaluation

Models were assessed using graphical methods, including the receiver operating characteristic (ROC) curve and the precision-recall curve (PRC), as well as numerical metrics, such as the AUROC, the area under the PRC curve (AUPRC), accuracy, Matthew's correlation coefficient (MCC), precision, and recall. To understand how the models made predictions, we employed SHapley Additive exPlanations (SHAP), a common machine learning visualization method. We used SHAP to determine the relative importance of predictor variables, in addition to performance plots and metrics.

## Online prediction tool

We have developed a web application (Fig 1) to predict individual patient outcomes based on their characteristics. The application is built using the models presented in our study. The source code and application can be accessed through Hugging Face, a platform that facilitates the sharing of ML models: https://huggingface.co/spaces/MSHS-Neurosurgery-Research/NSQIP-PCF. However, users must agree to the disclaimer we created based on other prediction tools before using our online tool [23].

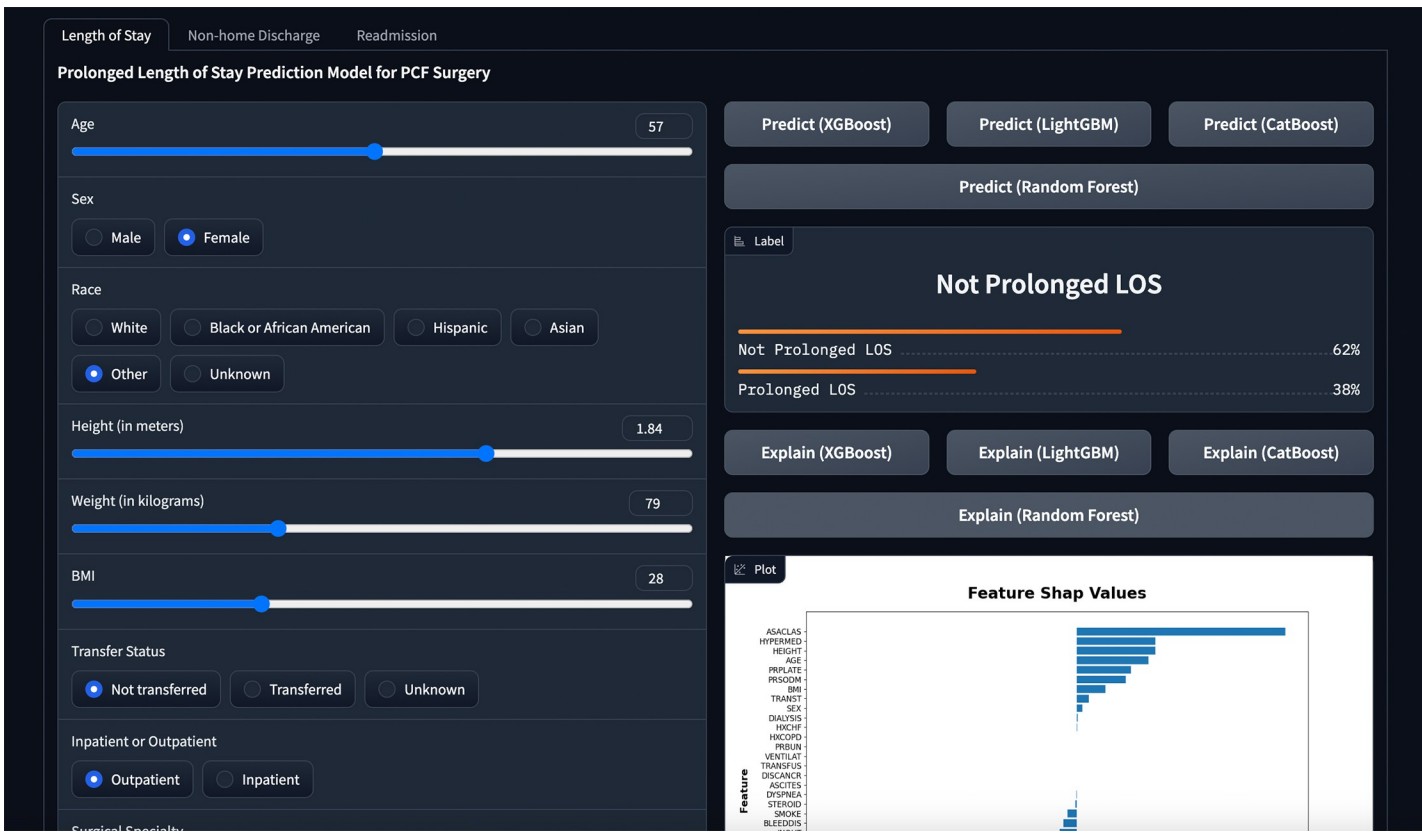

**Fig 1. A screenshot of the online web application.**

## Statistical analysis

Normally distributed continuous variables were reported as means (± standard deviations), while non-normally distributed continuous variables were presented as medians (interquartile ranges). The number of patients and their percentages were also reported for categorical variables. Group differences in outcomes were tested using different statistical tests depending on the type of variable. The independent t-test was used for normally distributed continuous variables with equal variances, and the Welch's t-test was used for normally distributed continuous variables with unequal variances. The Mann-Whitney U test was employed for non-normally distributed continuous variables, and the Pearson's chi-squared test was used for categorical variables. The normality of the variables was evaluated using the Shapiro-Wilk test, while Levene's test was used to assess the equality of variances for each variable. Statistical significance was determined at $p < 0.05$. The statistical analyses were conducted using Python version 3.7.15.

## Results

Initially, a total of 9843 patients were identified with the CPT code 22600. Inclusion and exclusion criteria were applied in a sequential manner. 1378 patients were excluded due to ICD codes, 647 due to CPT codes that were defined to be excluding, 1126 due to non-elective surgeries, 4 due to emergency surgeries, 16 due to anesthesia techniques other than general anesthesia, 22 due to surgical specialties other than neurosurgery or orthopedic surgery, 32 due to unclean wounds, 34 due to preoperative SIRS or sepsis, 278 due to ASA class 4, 5 or none assigned, 1 due to unknown LOS, and 28 due to discharge destination. 6277 patients were left in the analysis. There were 4807 patients with prolonged LOS, 1289 with non-home discharges, and 425 with readmissions. Characteristics of the patient population, both among the groups and in total, are presented in S3 and S4 Tables.

The most accurately predicted outcome in terms of AUROC was the non-home discharges, with a mean AUROC of 0.812 and an accuracy of 81.5%. The most accurately predicting algorithm in terms of AUROC was LightGBM, with a mean AUROC of 0.766, followed by CatBoost, with a mean AUROC of 0.764. The mean AUROCs for Random Forest and XGBoost were 0.763 and 0.752, respectively. Detailed metrics regarding the algorithms' performances are presented in Table 1. AUROC and AUPRC curves for the three outcomes are shown in Figs 2 and 3.

SHAP plots of the CatBoost model for the outcome prolonged LOS, the Random Forest model for the outcome non-home discharges, and the LightGBM model for the outcome readmissions are presented in Fig 4. The other SHAP plots can be seen in S1 Fig.

## Discussion

A group of ML models that can predict prolonged LOS, non-home discharges, and readmissions for patients who undergo PCF surgery is presented in this research. Our study suggests that the use of ML models could help with the risk stratification process for PCF surgery, and there is a substantial potential for predicting surgical outcomes. By providing patients with better information about the risks of surgery, clinicians may be able to customize patient care plans for those who are at risk of experiencing adverse outcomes following PCF surgery. This study contributes to the existing knowledge by describing the advantages and effectiveness of incorporating ML into patient care to anticipate outcomes after spine surgery [24].

The ML algorithms accurately predicted 74.6% to 78.5% of patients with prolonged LOS, exhibiting AUROC values ranging from 0.746 to 0.759. They also predicted 80.0% to 82.8% of patients with non-home discharges, with AUROC values between 0.804 and 0.819, and 93.6%

**Table 1. Metrics regarding the algorithms' performances.**

| | Algorithm | Precision | Recall | F1 | Accuracy | MCC | AUROC | AUPRC |
|---|---|---|---|---|---|---|---|---|
| **Length of Stay** | XGBoost | 0.370 | 0.521 | 0.433 | 0.764 | 0.296 | 0.732 | 0.504 |
| | LightGBM | 0.397 | **0.585** | 0.473 | **0.785** | 0.354 | 0.758 | 0.583 |
| | CatBoost | 0.485 | 0.540 | **0.511** | 0.775 | **0.366** | **0.759** | **0.594** |
| | Random Forest | **0.502** | 0.478 | 0.490 | 0.746 | 0.321 | 0.746 | 0.539 |
| | *Mean* | *0.439* | *0.531* | *0.477* | *0.768* | *0.334* | *0.749* | *0.555* |
| **Nonhome Discharge** | XGBoost | 0.516 | 0.608 | 0.558 | 0.816 | 0.446 | 0.804 | 0.599 |
| | LightGBM | 0.505 | **0.653** | **0.570** | **0.828** | **0.470** | 0.815 | 0.604 |
| | CatBoost | 0.527 | 0.611 | 0.565 | 0.818 | 0.453 | 0.811 | **0.640** |
| | Random Forest | **0.541** | 0.558 | 0.549 | 0.800 | 0.421 | **0.819** | 0.631 |
| | *Mean* | *0.522* | *0.608* | *0.561* | *0.815* | *0.447* | *0.812* | *0.618* |
| **Readmission** | XGBoost | 0.344 | 0.620 | 0.443 | 0.938 | 0.433 | 0.720 | 0.412 |
| | LightGBM | 0.356 | **0.727** | **0.478** | **0.944** | **0.484** | 0.724 | **0.456** |
| | CatBoost | 0.367 | 0.559 | 0.443 | 0.934 | 0.420 | 0.722 | 0.451 |
| | Random Forest | **0.378** | 0.576 | 0.456 | 0.936 | 0.434 | **0.724** | 0.435 |
| | *Mean* | *0.361* | *0.621* | *0.455* | *0.938* | *0.443* | *0.723* | *0.439* |

to 94.4% of patients with readmissions, featuring AUROC values from 0.720 to 0.724 in the test set. Based on average AUROC values across various outcomes, the LightGBM algorithm outperformed all other tested algorithms. These results indicate good classification performance for non-home discharge outcomes and fair performance for prolonged LOS and readmission outcomes [25].

A web application was developed and made publicly available worldwide. The web application allows users to access visual explanations of predictions made by four models that were employed in the study for the three outcomes of interest. The added interpretability of the predictions using SHAP values and figures could be valuable in clinical settings to optimize postoperative results for patients by addressing specific risk factors. As far as we know, this is the first ML-based web application that enables users to obtain predictions and explanations for postoperative outcomes following PCF surgery.

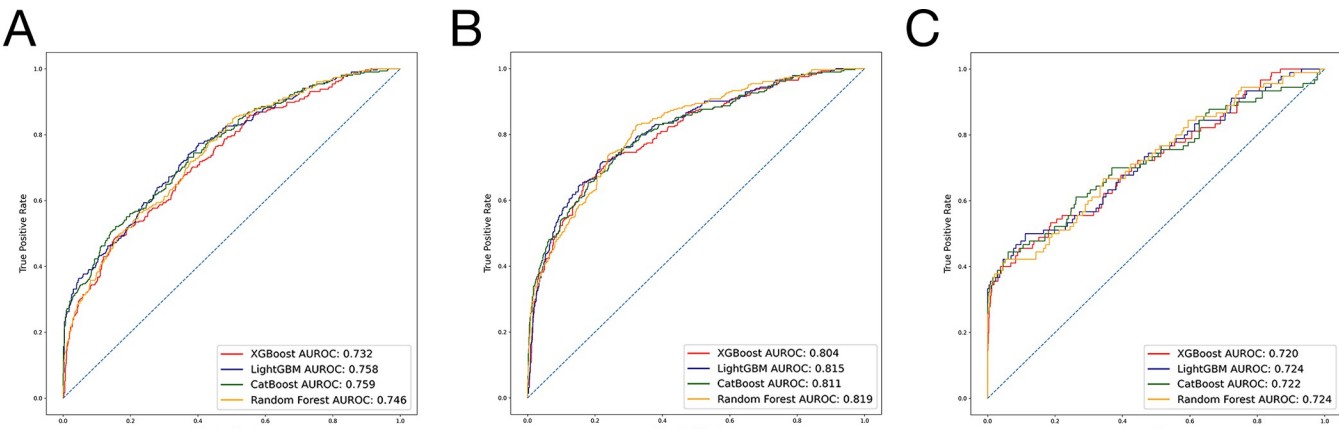

**Fig 2.** A. Algorithms' receiver operator curves for the outcome prolonged length of stay. B. Algorithms' receiver operator curves for the outcome nonhome discharges. C. Algorithms' receiver operator curves for the outcome readmissions.

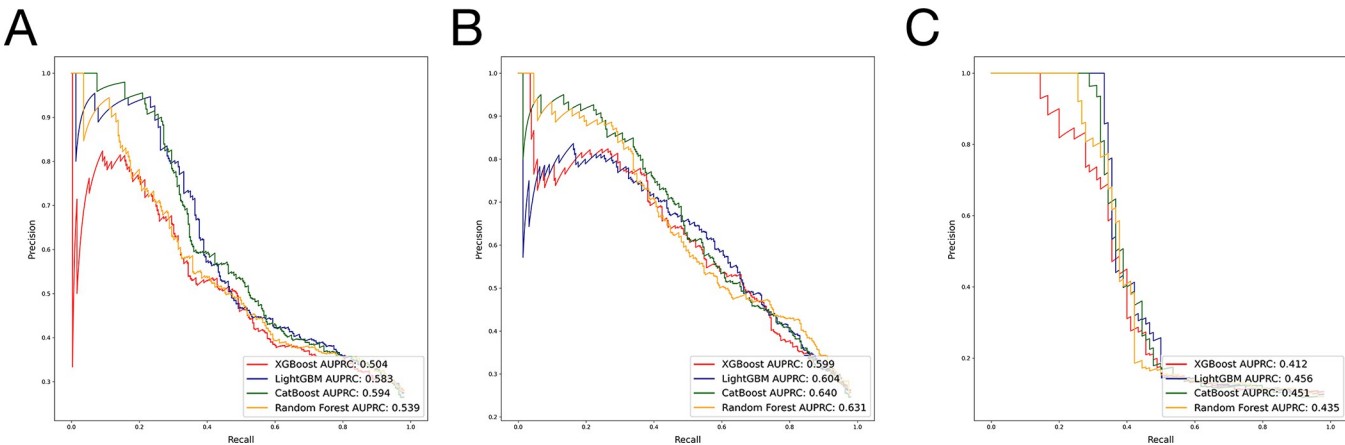

**Fig 3.** A. Algorithms' precision-recall curves for the outcome prolonged length of stay. B. Algorithms' precision-recall curves for the outcome nonhome discharges. C. Algorithms' precision-recall curves for the outcome readmissions.

Using different data sources, a few publications have analyzed the classification performance of ML algorithms in predicting postoperative outcomes upon PCF surgery. Cabrera et al., used random forest algorithm to predict short term postoperative outcomes following posterior cervical decompression with instrumented fusion [26]. The outcomes reoperation, readmission, LOS, transfusion requirement, and postoperative infection were predicted with AUROC values of 0.781, 0.791, 0.781, 0.902, and 0.724 respectively. In the study, the random forest algorithm identified a set of predictor variables for five outcomes of interest. The predictor variables included decreased preoperative hematocrit and white blood cell count, increased age, body mass index (BMI), operative time, and LOS, as well as postoperative infection. LOS was regarded as a predictive variable for models where LOS was not the desired outcome. Despite some studies using the NSQIP database that analyzed the accuracy of ML algorithms in predicting postoperative outcomes treated some of the variables that would not be known prior to the surgery as predictor variables, such as total operative time, length of stay and postoperative infection in the study by Cabrera et al., we did not adopt the same approach in our study. Instead of being the cause of undesirable outcomes, these postoperative variables might be among their consequences. These postoperative outcomes that were used as predictor variables were the most important predictors for every outcome that was investigated in the study. Using data that is not available at the time of prediction leads to inflated and ultimately false results, which is known as target leakage, a common pitfall in ML studies [27]. Shah et al. queried the California Office of Statewide Health and Planning and Development Patient Discharge Database from 2015 to 2017 to identify 6288 patients who underwent PCF [28]. Their primary outcome was readmission or major complication. The best-performing model had an AUROC of 0.679. Although the ensemble model identified novel prognostic features that were different from those most important for logistic regression, this study is limited by poor classification performances. Moreover, the reproducibility of the results is also limited because none of the aforementioned studies made the source code for the data preprocessing and classification models readily available to the readers. It was stated in the paper of Shah et al. that the code could be made available from the corresponding author on reasonable request. Finally, unlike our web application, which offers explanatory predictions for three different outcomes with four different algorithms, none of these studies provides a publicly accessible tool to be used by patients and clinicians.

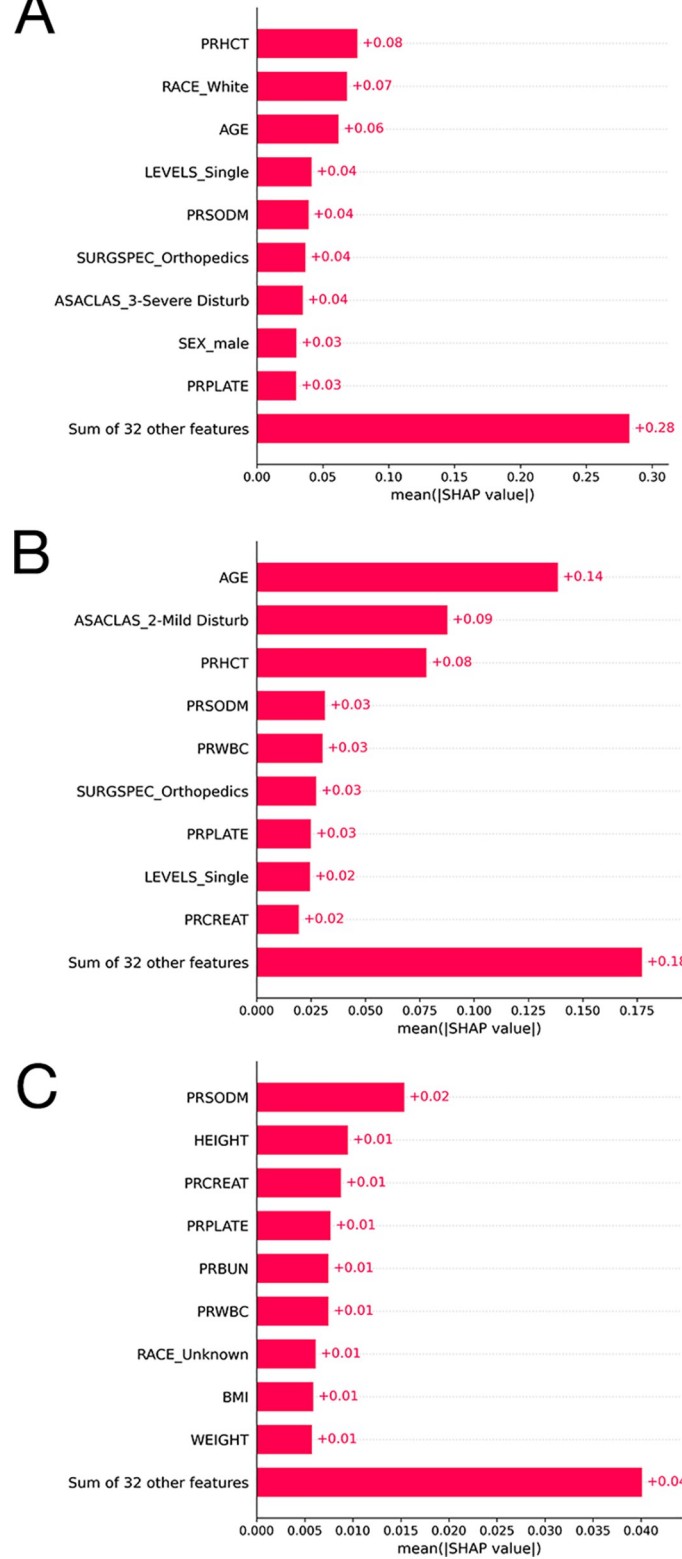

**Fig 4.** A. The ten most important features and their mean SHAP values for the model predicting prolonged length of stay with the CatBoost algorithm. B. The ten most important features and their mean SHAP values for the model predicting nonhome discharges with the Random Forest algorithm. C. The ten most important features and their mean SHAP values for the model predicting 30-day readmissions with the LightGBM algorithm.

The SHAP analysis results are in line with the most recent regression analysis-based research in terms of the relative significance of predictor variables. Shin et al. conducted a study with the NSQIP database to identify patient factors that are independently associated with prolonged LOS and readmission after PCF [29]. According to multivariate analysis, dependent functional status, diabetes mellitus, preoperative anemia, ASA class 3 or 4, and number of fused levels were significantly associated with prolonged LOS. Correspondingly, the best performing algorithm for prolonged LOS, CatBoost, had preoperative hematocrit, number of operated levels (multiple or single), and ASA class were among the most important ten features. In the same study, readmission was found to be significantly associated with dependent functional status and increased number of fused levels. However, these variables were not among the most important ten features utilized by our two best performing algorithms for readmission. Phan et al. found that anemic patients had higher rates of reoperation, unplanned readmission, and prolonged LOS following PCF [30]. Supporting these results, preoperative hematocrit was the most important feature for all four algorithms while predicting prolonged length of stay. Similarly, it was either the second or third most important feature for all four algorithms predicting nonhome discharges. Sridharan et al. used the NSQIP database to determine the impact of increased BMI on 30-day postoperative outcomes of PCF [31]. The authors concluded that increased BMI did not seem to have a major impact on 30-day postoperative outcomes following PCF, with the exception of a higher rate of deep surgical site infections seen in obese patients. On the other hand, BMI was among the top ten most important features by the two best performing algorithms for the outcome readmission.

Despite our robust methodology, our study has some potential limitations. Firstly, the sample of patients who underwent the PCF surgery may not have adequately represented all patients who underwent the procedure. The NSQIP dataset, which was used for our study, requires reporting from participating hospitals. Therefore, the sample of patients who underwent the PCF surgery may be overrepresented by patients from hospitals that have the infrastructure to meet NSQIP reporting requirements. Secondly, coding errors and inaccuracies that frequently occur in large clinical databases could have affected the study's results. Although the NSQIP database is frequently used, few studies have examined how accurate the coding actually is. Rolston et al. reported that CPT codes for neurosurgical procedures have many internal inconsistencies [32]. Additionally, the NSQIP data do not include specific factors that may be associated with a patient's risk of unfavorable postoperative outcomes. Although our current algorithm's mean AUROCs between 0.723 and 0.812 are considered fair to good classification performance, including other relevant variables and more granular data could improve the algorithm's performance.

## Conclusion

ML models hold significant potential in predicting postoperative outcomes after PCF surgery. These algorithms can be integrated into decision-making tools that have clinical relevance. The development and utilization of predictive models as easily accessible tools could substantially enhance prognosis and risk management. In this study, we introduce a predictive algorithm for PCF surgery with the goal of fulfilling the aforementioned objectives, which is available to the general public.

## Supporting information

**S1 Checklist. TRIPOD checklist: Prediction model development.**
(DOCX)

**S1 Fig.** A) The ten most important features and their mean SHAP values for the model predicting prolonged length of stay with the XGBoost algorithm, B) the ten most important features and their mean SHAP values for the model predicting prolonged length of stay with the LightGBM algorithm, C) the ten most important features and their mean SHAP values for the model predicting prolonged length of stay with the Random Forest algorithm, D) the ten most important features and their mean SHAP values for the model predicting nonhome discharges with the XGBoost algorithm, E) the ten most important features and their mean SHAP values for the model predicting nonhome discharges with the LightGBM algorithm, F) the ten most important features and their mean SHAP values for the model predicting nonhome discharges with the CatBoost algorithm, G) the ten most important features and their mean SHAP values for the model predicting readmissions with the XGBoost algorithm, H) the ten most important features and their mean SHAP values for the model predicting readmissions with the CatBoost algorithm, I) the ten most important features and their mean SHAP values for the model predicting readmissions with the Random Forest algorithm.
(TIF)

**S1 Table. CPT codes that were used to exclude patients.**
(DOCX)

**S2 Table. Characteristics of the patient population, both among the non-prolonged LOS and prolonged LOS groups and in total.**
(DOCX)

**S3 Table. Characteristics of the patient population, both among the home discharge and nonhome discharge groups and in total.**
(DOCX)

**S4 Table. Characteristics of the patient population, both among the no readmission and readmission groups and in total.**
(DOCX)

## Author Contributions

**Conceptualization:** Mert Karabacak, Konstantinos Margetis.

**Data curation:** Mert Karabacak.

**Formal analysis:** Mert Karabacak.

**Methodology:** Mert Karabacak, Konstantinos Margetis.

**Project administration:** Mert Karabacak.

**Software:** Mert Karabacak.

**Supervision:** Konstantinos Margetis.

**Visualization:** Mert Karabacak.

**Writing – original draft:** Mert Karabacak.

**Writing – review & editing:** Konstantinos Margetis.

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
