## [Decision Letter · Decision Letter 0]

5 Jun 2023

PONE-D-23-07187Interpretable machine learning models to predict short-term postoperative outcomes following posterior cervical fusionPLOS ONE

Dear Dr. Karabacak

Thank you for submitting your manuscript to PLOS ONE. After careful consideration, we feel that it has merit but does not fully meet PLOS ONE’s publication criteria as it currently stands. Therefore, we invite you to submit a revised version of the manuscript that addresses the points raised during the review process.

We look forward to receiving your revised manuscript.

Kind regards,

Mohamed El-Sayed Abdel-Wanis, Ph.D.

Academic Editor

PLOS ONE

Journal Requirements:

https://journals.plos.org/plosone/s/file?id=ba62/PLOSOne_formatting_sample_title_authors_affiliations.pdf 2. Please include your full ethics statement in the ‘Methods’ section of your manuscript file. In your statement, please include the full name of the IRB or ethics committee who approved or waived your study, as well as whether or not you obtained informed written or verbal consent. If consent was waived for your study, please include this information in your statement as well.

4. Please upload a new copy of Figures 1, 2 and 3 as the detail is not clear. Please follow the link for more information: " ext-link-type="uri" xlink:type="simple">https://blogs.plos.org/plos/2019/06/looking-good-tips-for-creating-your-plos-figures-graphics/"
https://blogs.plos.org/plos/2019/06/looking-good-tips-for-creating-your-plos-figures-graphics/

Additional Editor Comments :

The link

. https://huggingface.co/spaces/MSHS-Neurosurgery-Research/NSQIP-PCF.  

shows runtime error

. Line 61: What does the abbreviation “LOS” stand for? Meaning of any abbreviation should be mentioned on its first use, then the abbreviation should be used afterwards

. Line 116: “PT” repeated 2 times

. Line 117: meaning of the abbreviation “vs” should be mentioned on its first use “versus”

. Line 120: https://www.facs.org/quality-programs/data-and-registries/acs-nsqip/participant-use-data- file/).  ”File not found”

. Line 158: meaning of the abbreviation LOS (length of stay) should be mentioned on its first use (line 61).

. References

. Should be written as indexed in Index Medicus

Reviewers' comments:

Reviewer's Responses to Questions

**Comments to the Author**

1. Is the manuscript technically sound, and do the data support the conclusions?

Reviewer #1: Yes

Reviewer #2: Yes

2. Has the statistical analysis been performed appropriately and rigorously? 

Reviewer #1: Yes

Reviewer #2: I Don't Know

3. Have the authors made all data underlying the findings in their manuscript fully available?

Reviewer #1: Yes

Reviewer #2: Yes

4. Is the manuscript presented in an intelligible fashion and written in standard English?

Reviewer #1: Yes

Reviewer #2: Yes

5. Review Comments to the Author

Reviewer #1: I have read this interesting manuscript and have the following comments:

- The link 'https://huggingface.co/spaces/MSHS-Neurosurgery-Research/NSQIP-PCF' the authors placed is NOT functioning and always gives a Runtime Error !!!

- Why did the authors exclude patients with thoracic and lumbar fusions or operations for intraspinal lesions ??

- Why were Race and ethnicity variables were consolidated into one column, 'Race' ??

Reviewer #2: The authors use a registry based database and machine learning prediction algorithms to predict length of hospital stay, re-admissions and non-home discharge. They found that The most accurately predicted outcome in terms of the area under the receiver operating characteristic curve (AUROC) was the non-home discharges with a mean AUROC of 0.812, and the most accurately predicting algorithm in terms of AUROC was the LightGBM algorithm with a mean AUROC of 0.766.

The mentioned link to the prediction tool is still not working, i would prefer to submit a test version for the reviewers to be more able to test the tool.

The paper can be accepted for publication.

6. PLOS authors have the option to publish the peer review history of their article (what does this mean?). If published, this will include your full peer review and any attached files.

Reviewer #1: No

Reviewer #2: **Yes: **Ahmed Shawky Abdelgawaad

---

## [Author Response · Author response to Decision Letter 0]

5 Jun 2023

Response to Reviewers 

Journal Requirements

1. File naming was updated.

2. The ethics statement was moved to the ‘Methods’ section.

3. As previously described in the ‘Data availability’ section, restrictions apply to the availability of the data. We updated the ‘Data availability’ statement as follows: “Restrictions apply to the availability of these data, as the data is shared solely shared with fellows of American College of Surgeons. Data was obtained from American College of Surgeons National Surgical Quality Improvement Program and are available (https://www.facs.org/quality-programs/data-and-registries/acs-nsqip/) with the permission of American College of Surgeons.”

4. Figures were re-uploaded with better quality.

5. We confirm the accuracy of the references.

Additional Editor Comments

1. The application was updated and stabilized, therefore, the link should be working.

2. The meaning of the abbreviation ‘LOS’ is now mentioned in its first use.

3. PT is not used two times. The first one refers to International Normalized Ratio (INR) of prothrombin time (PT) values, and the second one refers to PT only.

4. The meaning of the abbreviation ‘vs.’ is now mentioned in its first use.

5. The link should be accessed without the parenthesis at the end and without any spaces: https://www.facs.org/quality-programs/data-and-registries/acs-nsqip/participant-use-data-file/

6. The meaning of the abbreviation ‘LOS’ is now mentioned in its first use.

7. We believe that our reference list is accurate, can you let us know specifically about any issues that you see?

Reviewer 1 Comments

1. The application was updated and stabilized, therefore, the link should be working.

2. We excluded these patients since patients undergoing posterior cervical fusion concomitantly with thoracic/lumbar fusion or resection of intraspinal lesions would have significantly different outcome profiles than patients undergoing posterior cervical fusion only.

3. We merged Race and Ethnicity columns in order to simplify the inputs for our web-application as much as possible.

Reviewer 2 Comments

1. The application was updated and stabilized, therefore, the link should be working.

---

## [Editor Report · Decision Letter 1]

26 Jun 2023

PONE-D-23-07187R1Interpretable machine learning models to predict short-term postoperative outcomes following posterior cervical fusionPLOS ONE

Dear Dr. Margetis,

Thank you for submitting your manuscript to PLOS ONE. After careful consideration, we feel that it has merit but does not fully meet PLOS ONE’s publication criteria as it currently stands. Therefore, we invite you to submit a revised version of the manuscript that addresses the points raised during the review process.

We look forward to receiving your revised manuscript.

Kind regards,

Mohamed El-Sayed Abdel-Wanis, Ph.D.

Academic Editor

PLOS ONE

Journal Requirements:

Additional Editor Comments:

Thank you for your reply

As regards the references:

Reference number 3: “Global Spine Journal” is indexed as Global Spine J.

Reference number 4: “World Neurosurgery” is indexed as “World Neurosurg”

Reference number 10: Journal of the American College of Surgeons is indexed as “J Am Coll Surg”

Please kindly review all references to confirm that all are reported as indexed in the Index medicus

Authors mentioned in their reply that figures were re-uploaded with better quality, however, I did not receive any figures in the revision file

---

## [Author Response · Author response to Decision Letter 1]

27 Jun 2023

Response to Reviewers 

Additional Editor Comments

References were updated as directed.

Figures were uploaded.

---

## [Editor Report · Decision Letter 2]

7 Jul 2023

Interpretable machine learning models to predict short-term postoperative outcomes following posterior cervical fusion

PONE-D-23-07187R2

Dear Dr. Konstantinos Margetis 

We’re pleased to inform you that your manuscript has been judged scientifically suitable for publication and will be formally accepted for publication once it meets all outstanding technical requirements.

Kind regards,

Mohamed El-Sayed Abdel-Wanis, Ph.D.

Academic Editor

PLOS ONE
---

## [Editor Report · Acceptance letter]

12 Jul 2023

PONE-D-23-07187R2 

Interpretable machine learning models to predict short-term postoperative outcomes following posterior cervical fusion 

Dear Dr. Margetis:

I'm pleased to inform you that your manuscript has been deemed suitable for publication in PLOS ONE. Congratulations! Your manuscript is now with our production department. 

Kind regards, 

on behalf of

Prof. Dr Mohamed El-Sayed Abdel-Wanis 

Academic Editor

PLOS ONE